# Total Nitrogen Shapes Diversity of Bloom-Forming Dinoflagellates in the Baltic Coastal Waters

**DOI:** 10.3390/biology15010048

**Published:** 2025-12-27

**Authors:** Irena V. Telesh, Hendrik Schubert, Sergei O. Skarlato

**Affiliations:** 1Zoological Institute of the Russian Academy of Sciences, St. Petersburg 199034, Russia; 2Institute for Biosciences, Faculty of Mathematics and Natural Sciences, The University of Rostock, 18051 Rostock, Germany; 3Institute of Cytology of the Russian Academy of Sciences, St. Petersburg 194064, Russia; sergei.skarlato@mail.ru

**Keywords:** Baltic Sea, dinoflagellates, eutrophication, harmful algal bloom, Intermediate Disturbance Hypothesis, mixotrophy, nitrogen, species richness

## Abstract

This study examined the diversity of dinoflagellates in coastal waters of the Baltic Sea at different eutrophication levels since many of these microplankton organisms form harmful algal blooms that deteriorate the environment. The analysis of long-term (44 years) database revealed altogether 82 dinoflagellate species; among them, ten species (seven mixotrophs and three heterotrophs) were most common, abundant, and regularly formed blooms. The discovered dinoflagellate diversity patterns correlated with eutrophication levels assessed by total nitrogen content in water. Maximal species richness was observed in the areas with the lowest nitrogen concentrations, i.e., at the smallest eutrophication. The findings were discussed in relation to the postulates of the Intermediate Disturbance Hypothesis. The results allowed updating the classification of eutrophication levels in the Baltic coastal waters, thus contributing to adequate assessment of the linkage between harmful dinoflagellate blooms and the process of eutrophication.

## 1. Introduction

The events of ample reproduction of algae in various water bodies worldwide have become rather frequent in recent decades, often resulting in large accumulations of phytoplankton biomass in the surface layers called “water blooms”, or harmful algal blooms (HABs) in the cases of toxic bloom aftereffects, or “red tides” in the cases of dinoflagellates’ blooms. After the termination of a bloom, the decomposition of these aggregates usually reduces the concentration of dissolved oxygen in water, thus deteriorating natural water quality and negatively affecting flora and fauna, fisheries and fish farming, aquaculture, tourism, and public health [1,2,3,4,5,6]. Despite a large array of available knowledge on HABs (reviewed in [7]), these devastating events are still hardly predictable so far. The reasons mainly rely on the fact that plankton dynamics depends on multiple inter-related and highly changeable external abiotic triggers and often poorly known internal biotic drivers [8,9,10,11,12,13,14,15,16].

Planktonic dinoflagellates represent a diverse group of protists that play key roles in aquatic biodiversity, primary productivity, and formation of blooms in marine and brackish waters around the globe. Meanwhile, inter- and intra-population heterogeneity, along with the ambiguity of dinoflagellates’ behavior under various stresses [10,17,18] and their peculiar life cycles [19,20,21,22,23], hinder the usage of the data on dinoflagellates for prognostic modeling of their HABs. Nevertheless, certain advances in unveiling some environmental prerequisites and fine cellular-molecular mechanisms underlying dinoflagellate blooms have been achieved in recent years [24]. In particular, several investigations of ecological niches and specific metabolism of these flagellates have contributed to HAB studies, since various trophic strategies of aquatic microorganisms are known to determine their ability to proliferate fast by modulating the growth rate and reproductive efficiency, providing competitive advantages and supporting the possibility of range expansion [9,25,26,27,28]. Specifically, many species of planktonic dinoflagellates, including those successfully invading new habitats, are capable of mixotrophic nutrition, i.e., they combine autotrophy due to photosynthesis with heterotrophy in the forms of osmotrophy and phagotrophy. Recently, an increasing number of not only dinoflagellates but also other microorganisms in phytoplankton, formerly known as “true autotrophs”, have been shown to feed mixotrophically [29,30,31,32]. Mixotrophic feeding mode boosts ecological fitness of planktonic organisms by enhancing their resistance to changing environment such as shifts in composition, amount, and proportions of nitrogen-containing compounds and other nutrients, or a decrease in illumination, which impedes photosynthesis [33,34,35].

Nitrogen-containing substrates are among the most essential nutrients that are indispensable for various cellular metabolic processes of living organisms used for their maintenance, growth, and reproduction [36]. In aquatic ecosystems, excess reactive nitrogen promotes the population growth of algae, which can trigger toxic blooms that deteriorate the environment, while nitrate in drinking water can harm human health [37]. Moreover, nitrogen is one of the few water pollutants, which is trending upwards nearly everywhere in the world, including in developed countries, despite strict regulation [38,39]. In recent decades, the global nitrogen cycle has been dramatically modified and intensified by various human activities, including agricultural practices, causing widespread impacts upon water quality [40]. The enhanced concentration of total nitrogen (TN) in water influences phytoplankton composition, abundance, and distribution; thus, TN is often considered an anthropogenic stress marker [37,41,42,43]. Meanwhile, the TN tolerance limits and optimum TN content for population growth of many common algal species, including a variety of bloom-forming dinoflagellates, are not known yet.

Recently, it was shown that dinoflagellates with the widest TN niche limits (along with broad niches for total phosphorus, water temperature and salinity) have the strongest potential for promoting most frequent bloom events in the fluctuating coastal environments [27]. The latter study proposed a conceptual model, which linked the dinoflagellates’ bloom-forming capacity with the width of their ecological niches, and allowed coming closer to prediction of “red tides”. However, some vital cellular mechanisms such as kleptoplastidy, or kleptoplasty—i.e., the usage of kleptoplasts (“stolen” chloroplasts of the consumed microalgae) as additional photosynthesis reactors by mixotrophs, which ensures their massive proliferation leading to blooms—have still not been investigated in dinoflagellates in sufficient detail [10,24,28,35,44]. In addition, the overall diversity of mixotrophic bloom-forming dinoflagellates has not been revealed yet, both regionally and globally, and the effects of TN content variation on dinoflagellates’ species richness likewise remain unclear.

To fill up some of the aforementioned knowledge gaps, this study aimed at demonstrating the diversity of the most common bloom-forming dinoflagellates and variations in their species richness in the southwestern Baltic Sea coastal waters under variable anthropogenic pressure, assessed using total nitrogen content in water. The study grounds on two premises. First, the enhanced TN concentration is known to promote eutrophication, shape structure and abundance of phytoplankton, and support algal blooms; therefore, it can be considered as a disturbance factor for diversity of planktonic algae [36]. Second, the Intermediate Disturbance Hypothesis (IDH; [45]) predicts maximum species richness in moderately TN-stressed waters. Therefore, based on these fundamental prerequisites, we used the extended dataset on physical–chemical characteristics and dinoflagellates diversity in the SW Baltic coastal waters to check the hypothesis suggesting that coastal waters with total nitrogen concentrations that are optimal to dinoflagellates might host greater taxonomic diversity of these protists compared to the areas with non-optimum TN content.

The research considers the ecological niche concept [46,47,48,49], sizes of TN niches of dominant dinoflagellates in the Baltic Sea [9,25,27], and current knowledge on flexibility of protistan mixotrophic feeding strategies [14,28,29,30]. The hypothesis was tested by the analysis of the long-term database, compiled during 44 years of research. The distribution of species richness of most common bloom-forming dinoflagellates in the SW Baltic coastal waters at different eutrophication levels was assessed, with special focus on mixotrophs. In addition, the updated classification of eutrophication levels in the Baltic coastal waters was developed.

## 2. Materials and Methods

For evaluation of the dinoflagellate diversity in plankton of the southwestern Baltic coastal waters, including the most common bloom-forming species of these protists, we performed the analysis of the long-term database described earlier by S. Sagert with co-authors [50], which was completed by incorporating the more recent data [9,25]. The amended database consists of altogether 7934 datasets containing information on phytoplankton taxonomic composition, abundance, and biomass, as well as the biotic and abiotic (physical–chemical) parameters sampled/measured at 64 stations located along the German Baltic Sea coast (Figure 1). The analyzed data cover an overall sampling period of 44 years, from 1972 to 2016.

Determination of basic physical (temperature, Secchi depth, pH) and chemical water properties (salinity and concentration of chlorophyll *a*, PO_4_, total phosphorus, TN, DIN, NO_2_, NO_3_, NH_4_, and SiO_4_) were carried out in parallel with phytoplankton sampling since these parameters are crucial for the development of algae. Total nitrogen and other abiotic characteristics were measured according to the COMBINE manuals of the HELCOM (see [50] and references therein for details).

The phytoplankton was sampled at 0.5 m below surface with a Niskin bottle sampler and an Apstein net (mesh size 100–150 µm), preserved in 4% formaldehyde and counted according to Utermöhl; algae biovolume was calculated by approximating the cell shape to simple geometrical figures [50]. As far as possible, taxonomic identifications were performed at the species level; samples with uncertain taxonomic identifications were not used in the analyses. More details of phytoplankton sampling, processing, determination of abiotic parameters, and the database characteristics are provided elsewhere [9,25,50].

The database was initially not uniform, as it was compiled using various sources: different monitoring programs, specific short- and mid-term field experiments, etc. Therefore, the datasets lacking any important parameters such as nutrient concentrations were excluded prior to the analysis. The removal of all datasets with incomplete information, and restriction to the data from the samples taken only between March and December (in order to exclude the periods of ice cover in January and February) resulted in 4534 datasets that were analyzed in this study.

The species-specific optimum concentrations of total nitrogen (TN opt, μmol/L) and niche limits of the top-10 most common bloom-forming dinoflagellate species in the SW Baltic coastal waters that formed the focus group of this study were considered according to our previously published data [27]. The TN niche width was calculated as the difference between the upper (max) and lower (min) niche limits. To analyze how the dinoflagellate species richness was changing within the TN gradient, we subdivided the optimal TN range for dinoflagellates (1–60 μmol/L) into five classes and calculated the mean number of dominant dinoflagellate taxa and the median TN concentrations for these TN classes, characterizing different eutrophication levels within the general TN niche limits of the studied species. The updated classification of eutrophication levels from low to extremely high was developed based on data published earlier [42,51].

Pearson correlation analysis was carried out for assessing the linkage between species-specific median TN concentrations and taxonomic diversity of dominant bloom-forming dinoflagellates in different areas of the SW Baltic coastal region. For all statistical tests, analyses, and visualization of the results, the program PRIMER V6 (Primer-E Ltd., Plymouth, UK) was used.

## 3. Results

### 3.1. Taxonomic Diversity of Dinoflagellates, Abundance, Dominant Species, and Their TN Niches

The analyses of a long-term database on phytoplankton and physical–chemical characteristics of the southwestern Baltic coastal waters revealed 82 dinoflagellate taxa at the genus and species levels inhabiting this region. The maximal registered TN concentration reached 659.8 μmol/L; however, the overall abundance of all dinoflagellates was the highest and exceeded 3300 cells/mL at TN concentrations below 50 μmol/L (Figure 2A). The same distribution pattern was revealed for the total number of dinoflagellate taxa (Figure 2B). Meanwhile, the majority of this diversity was registered in the areas where TN concentration in water was 8–70 μmol/L; the overall dinoflagellate taxonomic diversity therein peaked at a TN around 20 μmol/L, as shown by the distribution of the Shannon–Wiener diversity index (H′) within the TN range restricted to 8–70 μmol/L (Figure 2C).

Ten of the 82 dinoflagellate taxa were the most common species (Table 1) that regularly formed HABs in the SW Baltic coastal waters, as shown earlier [27]. Among these 10 selected dominants with species occurrences exceeding 100 in the total number of datasets (n) in the entire database, only three species are known as heterotrophs (*Amphidinium crassum*, *Protoperidinium steinii*, and *P. pellucidum*), while seven species from the genera *Prorocentrum*, *Dinophysis*, and *Ceratium* are mixotrophs (Table 1).

As shown in Table 1, the optimum TN concentration for the seven dominant bloom-forming mixotrophic dinoflagellates (MTDs) was in the range of 19.22–32.66 μmol/L. The heterotrophic dinoflagellates (HTDs) *Protoperidinium* spp. had comparable TN optimum values, while for the HTDs *Amphidinium crassum*, this parameter was extremely high reaching 146.9 μmol/L. The TN niche limits of mixotrophic dinoflagellates generally fell within the range of 8.69–55.59 μmol/L, except for one species, *Ceratium lineatum*, which had an exclusively low TN niche minimum (Table 1).

The TN niche width, calculated as the difference between upper (max) and lower (min) niche limits, for the dominant MTDs varied between 18.87 and 46.69 μmol/L. Meanwhile, for the HTDs, the TN niches were either significantly narrower (as for *Protoperidinium* spp.) or much broader (as for *Amphidinium crassum*), compared to the mixotrophs (Table 1). In addition, the mixotrophs were more diverse taxonomically compared to the heterotrophs. For the most common mixotrophic dinoflagellates in the SW Baltic coastal waters, the optimal TN niches were much wider, with maximal TN niche limits reaching up to twice as high as for the heterotrophic dinoflagellates (Figure 3).

The distribution of abundance of the selected species (*Prorocentrum cordatum*, *Ceratium lineatum*, and *Protoperidinium pellucidum*) in the gradient of TN values, with an indication of the species-specific optimum TN concentrations, is presented in Figure 4.

### 3.2. Species Richness of Dinoflagellates in the Gradient of TN Concentrations

The results showed that the overall TN niche limits for the selected most common dinoflagellate species in the SW Baltic coastal waters, excluding the extreme data for the heterotrophic *A. crassum*, roughly corresponded to the range of 1–60 μmol/L (Table 1). To analyze further how the dinoflagellate species richness changed within these TN limits, we subdivided the abovementioned overall TN range into five classes and calculated the mean number of dominant dinoflagellate taxa and the median TN concentrations for these TN classes characterizing different eutrophication levels within the general TN niche limits of the dinoflagellates (Table 2).

The results showed that coastal regions with total nitrogen concentrations corresponding to low eutrophication hosted nearly twice as high dinoflagellate diversity compared to waters with TN content typical for strong eutrophication (Table 2). Moreover, species richness of the dominant dinoflagellates in the Baltic coastal waters was the highest at low eutrophication conditions (TN < 15 µmol/L) and decreased gradually at moderate, high, and very high eutrophication, with the minimum number of species in the extremely eutrophic waters at TN > 45 µmol/L (Table 2).

The revealed trend was proven by a strong statistically significant correlation (R^2^ = 0.98, *p* ≤ 0.05) between the mean number of dinoflagellate taxa and the median total nitrogen concentration registered in each of the five TN classes characterizing different eutrophication levels within the TN niche limits of 1–60 µmol/L. Thus, our data did not support the theoretical maximum of the dinoflagellate species richness at the intermediate TN concentration around 30 µmol/L, as predicted by the IDH (Figure 5).

### 3.3. TN Content and Eutrophication Levels in Different Regions of the Baltic Sea

The data on TN concentrations in the SW Baltic coastal waters, collected and summarized in this study, allowed an update of the criteria for the classification of different eutrophication levels based on the previous classification [42]. In particular, in the late 1990s and the beginning of the XXI century, the annual average TN concentrations in different regions of the Baltic Sea varied significantly in space and time, ranging 17–376 µmol/L, with the maximum values reaching 659.8 µmol/L; however, the majority of data did not exceed 60 µmol/L (Table 3). Interestingly, this “critical” TN concentration of 60 µmol/L overlapped the upper TN limit for high occurrences of the selected most common mixotrophic dinoflagellate species in the SW Baltic coastal waters, as shown above in Table 1.

## 4. Discussion

Nitrogen pollution is one of the most important environmental issues of the XXI century [72]. Recent studies suggest that nitrogen may be the world’s largest global pressing externality due to its complex effects on the environment and human health [73]. Excessive nitrogen loading of water bodies can create adverse environmental effects on aquatic ecosystems through three biochemical mechanisms [74]. Those are eutrophication, acidification, and direct toxicity; all these processes might cause numerous problems, such as proliferation of HABs, exacerbation of hypoxic zones, fish mortality, and loss of biodiversity [7,75,76,77]. Moreover, the world has also surpassed the planetary boundary for nitrogen—a level of human interference beyond which environmental damage increases dramatically and possibly permanently [78]. Thus, understanding the impact of nitrogen content on the diversity and functions of biota in marine coastal ecosystems, particularly those subjected to eutrophication and harmful blooms of microorganisms, including toxic or potentially toxic algal species, is a topical issue of current aquatic ecology [10,28,47].

In this study, we demonstrated how variable anthropogenic pressure, assessed using data on total nitrogen content in water, affected the variations in species richness of bloom-forming dinoflagellates, and updated the classification criteria for eutrophication levels of the Baltic Sea waters. Nearly two decades ago, in his classification, U. Schiewer [42] suggested that the TN concentration ranging 17.8–22.1 µmol/L corresponded to “small or moderate eutrophication effect”, 22.1–25.7 µmol/L corresponded to “severe” effect, and TN > 25.7 µmol/L corresponded to “very serious eutrophication effect”. This classification, based on the criteria for ascertaining the eutrophication effects by changes in the major environmental parameters according to the conceptual flow-models [41], covers a rather narrow range of TN values. Therefore, according to this classification, the majority of Baltic coastal regions were suffering from “very serious” eutrophication effects, except for the Askö Area, Bothnian Sea, Bothnian Bay, and Kattegat, as shown in Table 3. Since then, however, intensifying human activities have imposed immense challenges in sustainable water quality management due to enhanced eutrophication of the Baltic Sea [79,80]. Importantly, eutrophication influenced the cause-effect interplay of biotic drivers and physical–chemical triggers of HABs, affecting the bloom-forming potential of algae in coastal and estuarine environments [9]. Therefore, here we modified the above classification based on the recent knowledge of the TN content in the Baltic coastal waters and linked those data to the TN niche sizes of the most common bloom-forming dinoflagellates. According to the updated classification, low to moderate eutrophication levels correspond to TN concentrations below 25 µmol/L, and extremely high TN content correspond to above 45 µmol/L (Table 2). This modified classification covers a wider range of TN content values compared to classification by Schiewer [42] and is consistent with the knowledge on growing eutrophication of the Baltic Sea [79,80,81]. However, the data in Table 3 show that, irrespective of the fact that the new classification is more detailed, there is a good consensus between the basic eutrophication levels in different Baltic Sea regions estimated by both classifications.

Interestingly, the results of our study of spatial distribution of the dinoflagellate species richness in the SW Baltic coastal waters, assessed using the 44 year-long database described in detail earlier [9,25,50], did not support our main research hypothesis. The latter supposed that, according to the IDH, those coastal waters with total nitrogen concentrations that were optimal for bloom-forming dinoflagellates hosted greater taxonomic diversity of these protists compared to the areas with non-optimum TN content. However, the results of the current study showed that waters with TN concentrations that corresponded to low eutrophication hosted nearly twice as high dinoflagellate diversity compared to waters with TN values typical for strong eutrophication (as shown in Figure 5). Moreover, these results contradict the Intermediate Disturbance Hypothesis, which postulates that maximum species richness should be expected at moderately stressed conditions [45]. Meanwhile, we discovered that the highest species richness of dominant bloom-forming dinoflagellates was associated with TN concentrations that were much lower than the optimum TN content for these species, as measured earlier during the assessment of TN niches [27].

The reasons for this peculiar outcome may be multiple, for example, the interfering stressful (for biodiversity) influences of other environmental factors and their combinations [9,10], or the repeatedly debated controversy of the IDH [82]. Meanwhile, here we suggest another presumable explanation of the obtained results, based on both laboratory experiments and field observations, which still supports the IDH, although this explanation is largely speculative. Specifically, as shown by experimental approaches [83,84], the increased species diversity depends on the frequency of disturbances rather than the amplitude of their fluctuations. A diversity maximum is observed usually when disturbance frequencies are equal to a three-generation cycle [85]. For dinoflagellates at optimum (i.e., nutrient-saturated) conditions, the growth rates between ca. 0.3–0.7 day^−1^ are reported from laboratory experiments (see [86] for an overview), and this is in good accordance with field observations [57]. At suboptimal conditions, the growth rates of algae are decreasing. Consequently, this will shift the optimum disturbance pattern from about four to ten days to much longer periods of time, allowing for variability of other factors than TN to serve as disturbances, affecting the species’ competition pattern. Slower growth of the superior competitor, in turn, allows for the increased survival period of the inferior ones. Summarizing, whereas at optimum conditions fast and rhythmic changes in main determinants are required for the IDH-supported diversity effects, at suboptimal conditions, the competition rate is dampened by slower growth rates. Thus, the chance that other factors than TN would also serve as disturbances is increasing, and this is leading to higher taxonomic diversity.

Similarly to our results, the reduction in species richness in the environments with increased nutrient availability has been observed for phytoplankton, zooplankton, and mycoplankton [87,88,89]. For the cyanobacteria whose growth rates exceed those of eukaryotes [90], and for aquatic macro-organisms with annual reproduction cycle and slower growth rates [91], the picture looks different, which also could be considered as a pro-argument for the possible (still speculative) explanation of the inconsistency of our results for dinoflagellates and the IDH.

Moreover, we admit that in the study region, the dominance of mixotrophic dinoflagellates with wide ecological niches, including the TN niches that provide grounds for successful coexistence of many dinoflagellate species at relatively low TN concentrations, is one of the major reasons ensuring the discovered trend. Indeed, since the majority of dominant dinoflagellates in the studied area are mixotrophs, the maximal number of dinoflagellate species at low eutrophication conditions, and the subsequent gradual decrease in species richness with the increasing TN concentration, could be the outcome of the effective partitioning of trophic niches by mixotrophic dinoflagellates [24,25]. The latter are known to effectively consume various nitrogen-containing substrates in different proportions [35,92,93] and express high inter- and intra-population heterogeneity along with the ambiguity of dinoflagellates’ behavior under various external stresses [10,17,18].

Thus, unveiling the diversity of bloom-forming dinoflagellates and variations in their species richness within the gradient of multiple anthropogenic pressures, assessed using total nitrogen content in water, is of extreme importance for understanding eutrophication and forecasting HABs. Future analyses could possibly embrace databases compiled at larger geographical scales (e.g., the whole Baltic Sea or several European seas). This could help overcome certain limitations of the current study, for example, a relatively low number of the analyzed most common bloom-forming dinoflagellates (10 species) in the region, which might hamper some statistical analyses. Other possible limitations should be considered as topical issues for future research, such as the assessment of the dinoflagellate diversity fluctuations within a wider range of TN concentrations, compared to the present study, or the search for a more solid argumentation for the divergence of the currently obtained data from the distribution patterns predicted by the Intermediate Disturbance Hypothesis.

## 5. Conclusions

In total, 82 dinoflagellate species were shown to inhabit the SW Baltic coastal waters; ten of those were the most common and abundant bloom-forming algae. Our research hypothesis, suggesting those coastal waters with total nitrogen concentrations that are optimal for bloom-forming dinoflagellates to host greater taxonomic diversity of these protists compared to the areas with non-optimum TN content, was not supported by the analyses of the 44 year-long database from the SW Baltic coastal waters. We observed the highest species richness of dinoflagellates at relatively low eutrophication level (TN < 15 µmol/L). Moreover, the diversity of dominant bloom-forming dinoflagellate species decreased gradually from low to moderate, to high and very high eutrophication levels, with the minimum number of species in the extremely eutrophic waters at TN > 45 µmol/L. These results disagree with the theoretical species-richness maximum at the intermediate TN concentrations (around 30 µmol/L), as predicted by the Intermediate Disturbance Hypothesis. A possible (although speculative) explanation of this discrepancy could involve the linkage between the dinoflagellate growth rates and TN disturbance frequencies (rather than disturbance magnitude). The relation between these parameters secures different competition patterns at the optimal and suboptimal TN conditions, promoting higher dinoflagellate species richness at suboptimal TN concentrations due to lower growth rates and, therefore, less effective competition for nutrients. In addition, the current results allowed updating the classification of eutrophication levels in the Baltic Sea coastal waters based on total nitrogen content linked to the diversity of bloom-forming dinoflagellates at various eutrophication conditions. The results of the study can contribute to mathematical modeling of HABs dynamics under growing eutrophication and support the development of comprehensive HAB control strategies for environmental management aimed at effective protection of natural aquatic ecosystems.

## Figures and Tables

**Figure 1 biology-15-00048-f001:**
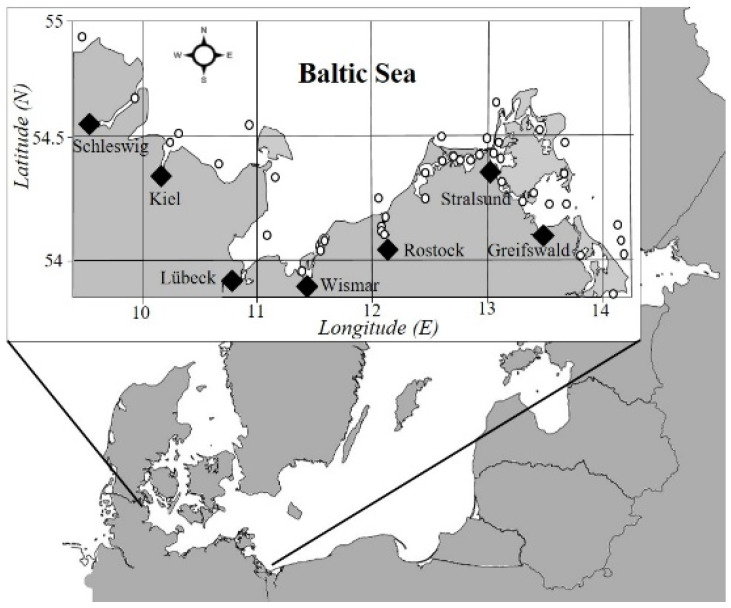
Schematic map of the coastal study region in the southwestern Baltic Sea. Empty circles indicate the main sampling stations represented in the database and described in detail earlier [9,25].

**Figure 2 biology-15-00048-f002:**
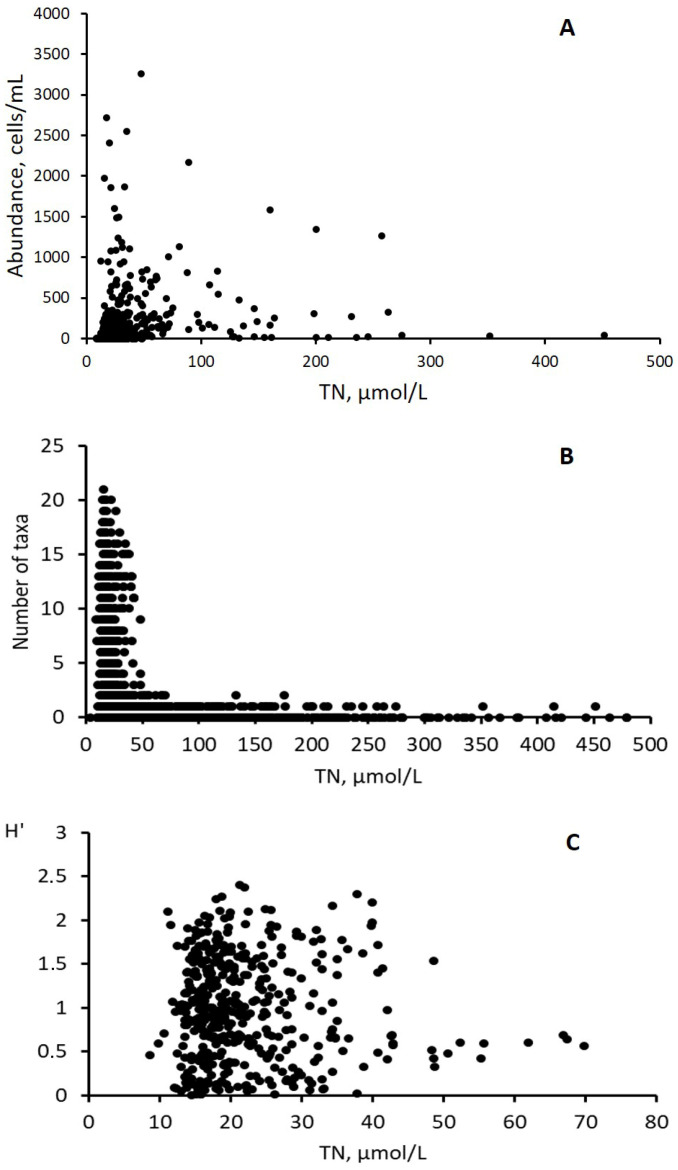
Abundance and diversity of dinoflagellates in the gradient of total nitrogen concentrations (TN, μmol/L) in the SW Baltic coastal waters. (**A**) Total abundance (cells/mL) of all 10 dominant dinoflagellate species; (**B**) number of all dinoflagellate taxa; (**C**) Shannon–Wiener diversity index (H′) for all dinoflagellate species registered in the waters with TN concentrations of 8–70 μmol/L.

**Figure 3 biology-15-00048-f003:**
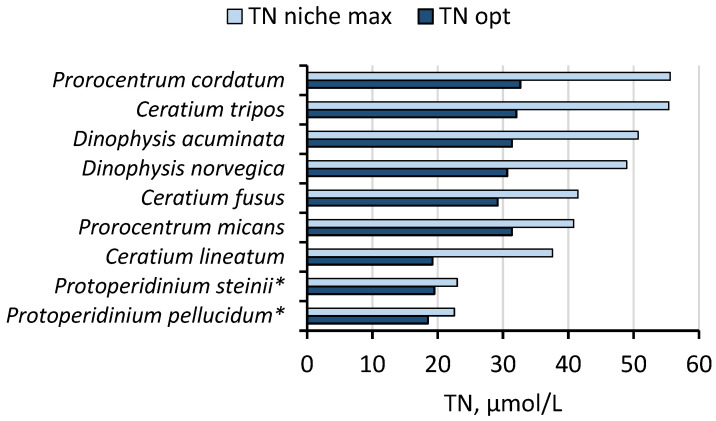
The optimal TN niches (TN opt, µmol/L) and maximal TN niche limits (TN niche max, µmol/L) of the most common bloom-forming dinoflagellates in the SW Baltic coastal waters. For TN niche-range values see Table 1. Asterisks mark heterotrophic dinoflagellate species (HTDs). *A. crassum* was excluded because of its extremely high maximum TN niche values to improve visibility of the results for the majority of studied species.

**Figure 4 biology-15-00048-f004:**
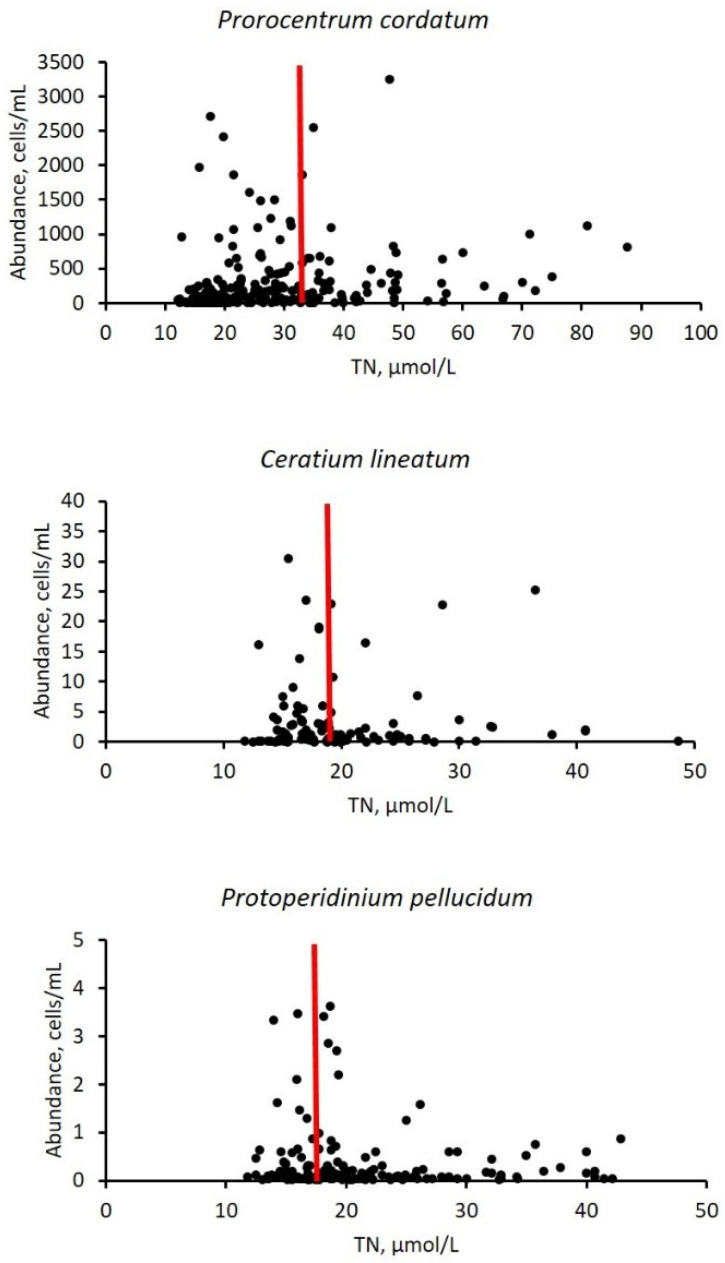
Distribution of abundance (cells/mL) of selected species (*Prorocentrum cordatum, Ceratium lineatum*, and *Protoperidinium pellucidum*) in the gradient of TN values (μmol/L; restricted axis). Red vertical bars indicate species-specific optimum TN concentrations.

**Figure 5 biology-15-00048-f005:**
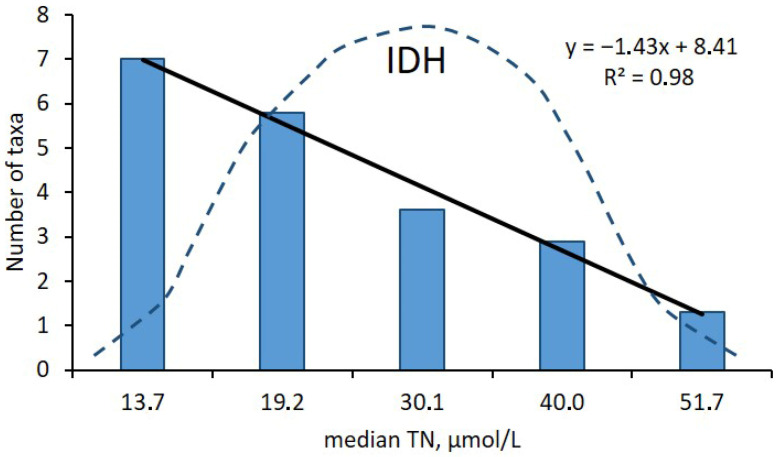
Correlation (R^2^ = 0.98; *p* ≤ 0.05; black solid line) of the mean number of dinoflagellate taxa (bars) with the median total nitrogen concentration (TN, µmol/L) registered in each of the five TN classes characterizing different eutrophication levels within the TN niche limits of 1–60 µmol/L in the SW Baltic coastal waters. Broken line indicates the hypothetical distribution of the number of dinoflagellate taxa within the TN gradient according to the Intermediate Disturbance Hypothesis (IDH).

**Table 1 biology-15-00048-t001:** The optimum concentration of total nitrogen (TN opt, μmol/L), niche limits, and niche width of the top-10 most common bloom-forming dinoflagellate species in the SW Baltic coastal waters (according to data from [27]); n—total number of the datasets with the species’ occurrences >100 in the database. Dinoflagellates nutrition modes are indicated according to published data. HTD—Heterotrophic species (marked by asterisk); MTD—mixotrophic species (in bold).

Species	n	TN opt (μmol/L)	TN Niche Limits, Min–Max (μmol/L)	TN Niche Width (μmol/L)	Nutrition Mode [Reference]
*Amphidinium crassum **	115	146.90	91.12–202.69	111.57	HTD [52,53]
*Protoperidinium pellucidum **	165	18.52	14.48–22.55	8.07	HTD [54]
*Protoperidinium steinii **	107	19.50	15.31–23.0	7.69	HTD [55,56]
** *Ceratium fusus* **	253	**29.19**	16.92–41.47	24.55	**MTD** [57]
** *Ceratium lineatum* **	108	**19.22**	0.87–37.57	36.70	**MTD** [58]
** *Ceratium tripos* **	282	**32.04**	8.69–55.38	46.69	**MTD** [58]
** *Dinophysis acuminata* **	197	**31.38**	12.07–50.68	38.61	**MTD** [59]
** *Dinophysis norvegica* **	232	**30.65**	12.37–48.92	36.55	**MTD** [60]
** *Prorocentrum cordatum* **	457	**32.66**	9.74–55.59	45.85	**MTD** [61]
** *Prorocentrum micans* **	242	**31.38**	21.95–40.82	18.87	**MTD** [61]

**Table 2 biology-15-00048-t002:** TN cut-off thresholds for eutrophication levels, mean number of dinoflagellate taxa, and median TN concentrations calculated for the five TN classes characterizing different eutrophication within the general dinoflagellate TN niche limits (1–60 µmol/L). The classification of five eutrophication levels, from low (level 1) to extremely high (level 5), was developed based on data in [42,51], with modification. SD—standard deviation; n—the number of datasets analyzed for each TN class.

TN Range Classes	TN Cut-Off Thresholds (µmol/L)	Eutrophication Level	TN Median (µmol/L)	Number of Taxa	±SD	n
1	1–15	Low (1)	13.7	7.0	5.1	82
2	16–25	Moderate (2)	19.2	5.8	5.4	428
3	26–35	High (3)	30.1	3.6	4.3	180
4	36–45	Very high (4)	40.0	2.9	4.0	77
5	46–60	Extremely high (5)	51.7	1.3	1.2	58

**Table 3 biology-15-00048-t003:** Annual average (and/or mean, median, range, or maximum) total nitrogen content (TN, μmol/L) and eutrophication levels (1 to 5; see Table 2 for TN cut-off thresholds) in different regions of the SW Baltic Sea according to earlier classification [42] and current assessment [this study].

Region, Year	TN, μmol/L [Source]	Eutrophication Level After U. Schiewer [42]/This Study
South-western (German) Baltic coastal waters	32.9 (median), 57.3 (mean), SD 65.1, range 0.18–659.8 [this study]	very serious/**high (3) to extremely high (5)**
Kattegat	22 ± 3 [51]	moderate/**moderate (2)**
Baltic Sea proper	22 ± 3 [51]	moderate/**moderate (2)**
Darss-Zingster Bodden, 1996–2004	205 (138–253) [62]	very serious/**extremely high (5)**
Greifswalder Bodden	376 [63]	very serious/**extremely high (5)**
Gulf of Gdańsk	26.96 [64]	very serious/**high (3)**
Vistula Lagoon	20–400 [65]	very serious/**extremely high (5)**
Curonian Lagoon	119.7 (15–453.2) [66]	very serious/**extremely high (5)**
Gulf of Riga	39 ± 11 [51]	very serious/**very high (4)**
Gulf of Finland	30 ± 3 [51]	very serious/**high (3)**
Eastern Gulf of Finland, 1990s	23.5–31.1 [67]	very serious/**high (3)**
Eastern Gulf of Finland (deep), 2022	31.5 (max 42.19) [68]	very serious/**high (3)**
Eastern Gulf of Finland (shallow), 2022	41.27 (max 97.74) [68]	very serious/**very high (4)**
Neva Bay, 1990s	24.9–30.4 [67]	very serious/**high (3)**
Neva Bay, 2022	45.32 (max 148.6) [68]	very serious/**very high (4)**
Stockholm Archipelago	60 [69]	very serious/**extremely high (5)**
Askö Area	17 [70]	small/**moderate (2)**
Odense Fjord, 1979–2003	23–115 [71]	very serious/**extremely high (5)**
Bothnian Sea	19 ± 2 [51]	small or moderate/**moderate (2)**
Bothnian Bay	20 ± 2 [51]	small or moderate/**moderate (2)**

## Data Availability

The datasets used and/or analyzed during the current study are available from the corresponding authors (I.V.T. and H.S.) on reasonable request.

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
