# Peer review of "Biology2026, 15(1), 48;https://doi.org/10.3390/biology15010048"

_biology, 2025, doi:10.3390/biology15010048_

Round 1

Reviewer 1 Report

Comments and Suggestions for Authors

This study by Telesh and colleagues performs a meta-analysis of 44 years of data to be able to study the effects of total nitrogen on dinoflagellate blooms in the coastal region of the Baltic Sea. Methods and statistics used are appropriate for the study design. The key findings of the study are that at least species richness occurs at low levels of eutrophication and that it is claimed that species richness follows the intermediate disturbance hypothesis. This article fits with the readership of Biology. Below are recommendations on how to improve the manuscript:

Introduction:

-The first two sentences can be combined for clarity. (lines 47-52)

-line 67 - new data should be recent studies or several studies

-the sentence in lines 114-116 needs to be rewritten due to repetitive wording. That paragraph should be further expanded given its importance to the paper.

Results: 

-Please improve the quality of figures 2 and 4. 

-Please add error bars to figure 3.

- Figure 5 seems to disagree with the IDH? Are the data that do follow the IDH? This needs to be addressed as it directly conflicts with the conclusions. 

- Table 2 - make sure to format tables the same way. This one is missing the bottom grid line

- I would recommend adding a table or a figure indicating the cut off thresholds for eutrophication levels indicated in Table 3.

Author Response

Dear Reviewer 1,

Thank you very much for your valuable comments and recommendations that allowed us to improve the manuscript substantially. Below, please find our “step-by-step” responses to each of your comments and suggestions.

With sincere gratitude,

The authors

Comments and Suggestions for Authors

This study by Telesh and colleagues performs a meta-analysis of 44 years of data to be able to study the effects of total nitrogen on dinoflagellate blooms in the coastal region of the Baltic Sea. Methods and statistics used are appropriate for the study design. The key findings of the study are that at least species richness occurs at low levels of eutrophication and that it is claimed that species richness follows the intermediate disturbance hypothesis. This article fits with the readership of Biology. Below are recommendations on how to improve the manuscript:

Introduction:

COMMENT

-The first two sentences can be combined for clarity. (lines 47-52)

RESPONSE

Following the Reviewer’s recommendation, we combined the first two sentences of the Introduction in one, and shortened it for clarity. The revised first phrase of the Introduction is now as follows:

“The events of ample reproduction of algae in various water bodies worldwide have become rather frequent in recent decades, often resulting in large accumulations of phytoplankton biomass in the surface layers called "water blooms", or harmful algal blooms (HABs) in the cases of toxic bloom aftereffects, or “red tides” in the cases of dinoflagellates’ blooms”.

COMMENT

-line 67 - new data should be recent studies or several studies

RESPONSE

Corrected to “several investigations”.

COMMENT

-the sentence in lines 114-116 needs to be rewritten due to repetitive wording. That paragraph should be further expanded given its importance to the paper.

RESPONSE

To avoid repetitive wording, the sentence was rewritten as follows: “Second, the Intermediate Disturbance Hypothesis (IDH) predicts maximum species richness in moderately TN-stressed waters [45]”. We also expanded the paragraph given its importance to the paper and provided additional explanations of our research hypothesis and methods used for its verification (lines 116–122).

Results:

COMMENT

-Please improve the quality of figures 2 and 4.

RESPONSE

The quality of Figures 2 and 4 was improved. New versions of the figures were inserted in the text and additionally submitted as separate files.

COMMENT

-Please add error bars to figure 3.

RESPONSE

The TN niche-range values that demonstrate the variability of TN niches of the studied dinoflagellates are given in Table 1, which precedes Figure 3. Therefore, in Figure 3, we do not show error bars nor range values in order to avoid duplication of the information and elude overloading the figure. The aim of this figure is to express most clearly the main results – the optimum and maximum TN niche values (but not the overall niche sizes) of the dominant bloom-forming dinoflagellates in the SW Baltic coastal waters. Earlier, we published a separate paper on the dinoflagellate niche sizes in the Baltic Sea (please, see: Telesh I., Schubert H., Skarlato S., 2024. Wide ecological niches ensure frequent harmful dinoflagellate blooms. HELIYON, 10 (4): e26495. https://doi.org/10.1016/j.heliyon.2024.e26495). Thus, here we improved the data presentation by simply adding a clarifying sentence to the legend of Figure 3: “For TN niche-range values see Table 1”.

COMMENT

- Figure 5 seems to disagree with the IDH? Are the data that do follow the IDH? This needs to be addressed as it directly conflicts with the conclusions.

RESPONSE

Figure 5 shows that our data does not support the theoretical maximum of the dinoflagellate species richness at the intermediate TN concentration around 30 µmol/L, as predicted by the IDH. In our study, no data on dinoflagellate diversity within the studied TN range (from 1 to 60 µmol/L) follow the IDH. This is our major conclusion, which does not conflict with the data. On the contrary, this conclusion is fully supported by a very large set of our data on dinoflagellates and TN values obtained within long-term studies during more than 40 years. Specifically, we discovered that the highest species richness of dominant bloom-forming dinoflagellates was associated with TN concentrations that were much lower than the optimum TN content for these species, which disagrees with IDH. Similar results were obtained also by other authors for phytoplankton, zooplankton and mycoplankton [references 87-89]. In the Discussion section, we analysed these results and discussed in detail possible reasons of the discovered phenomenon and the described biodiversity pattern (please, see pages 10–12 in the revised manuscript).

COMMENT

- Table 2 - make sure to format tables the same way. This one is missing the bottom grid line

RESPONSE

Corrected (the bottom grid line was added).

COMMENT

- I would recommend adding a table or a figure indicating the cut off thresholds for eutrophication levels indicated in Table 3.

RESPONSE

Thank you very much for the comment. The cut off thresholds for eutrophication levels mentioned in Table 3 were initially given in Table 2, column “TN range classes”. In the revised version, we corrected the name of this column in Table 2 to “TN cut off thresholds” and provided additional explanations in the headings of both tables, 2 and 3, to clarify this issue.

Reviewer 2 Report

Comments and Suggestions for Authors

The manuscript entitled "Total Nitrogen Shapes Diversity of Bloom-Forming Dinoflagellates in the Baltic Coastal Waters" is devoted to the impact of nitrogen on harmful algal blooms (HABs) and the functions of biota in marine ecosystems of the Baltic Sea under eutrophication. The authors analyzed a 44-year-long database, identifying 82 dinoflagellate species. The results showed that the highest dinoflagellate species richness was associated with much lower TN concentrations than the optimum values ​​for these species. In my opinion, this manuscript is consistent with the aims and scope of Molecules Journal. This is a rare case where the article is, in my opinion, practically flawed. I am ready to recommend it for publication, due to the few comments below.
1. It is unclear whether concentrations of nitrogen species were measured during sampling in this study. If so, the method used should be indicated. In addition, the sampling should include the main physical, chemical, and geochemical parameters that may have an important role in phytoplankton development.
2. The research methods section should be structured in more detail, specifying measurement methods at various points.

Author Response

Dear Reviewer 2,

Thank you very much for your valuable comments that allowed us to improve substantially the presentation of methodology used in our study. Below, please find our responses to your comments and suggestions.

With sincere gratitude,

The authors

Comments and Suggestions for Authors

The manuscript entitled "Total Nitrogen Shapes Diversity of Bloom-Forming Dinoflagellates in the Baltic Coastal Waters" is devoted to the impact of nitrogen on harmful algal blooms (HABs) and the functions of biota in marine ecosystems of the Baltic Sea under eutrophication. The authors analyzed a 44-year-long database, identifying 82 dinoflagellate species. The results showed that the highest dinoflagellate species richness was associated with much lower TN concentrations than the optimum values ​​for these species. In my opinion, this manuscript is consistent with the aims and scope of Molecules Journal. This is a rare case where the article is, in my opinion, practically flawed. I am ready to recommend it for publication, due to the few comments below.

COMMENT

  1. It is unclear whether concentrations of nitrogen species were measured during sampling in this study. If so, the method used should be indicated. In addition, the sampling should include the main physical, chemical, and geochemical parameters that may have an important role in phytoplankton development.

RESPONSE

In this study, we used the database for which the determination of basic physical (temperature, Secchi depth, pH) and chemical water properties (salinity, Chl-a, PO4, TP, TN, DIN, NO2, NO3, NH4, and SiO4) were carried out in parallel with phytoplankton sampling since these parameters are crucial for the development of algae. This information was added on page 4 of the revised manuscript.

However, in this study we analyzed only dinoflagellates and the TN concentration in water. Therefore, the measurement details for other water properties were not given in the text of the manuscript.

Meanwhile, our results analyzing other abovementioned parameters were published earlier, in several previous articles that are cited in this manuscript; for example, please see: Telesh, I.; Schubert, H.; Skarlato, S. Wide ecological niches ensure frequent harmful dinoflagellate blooms. Heliyon 2024, 10 (4): e26495. https://doi.org/10.1016/j.heliyon.2024.e26495

COMMENT

  1. The research methods section should be structured in more detail, specifying measurement methods at various points.

RESPONSE

We improved the structure of the section Materials and Methods by adding the information about the main physical and chemical parameters that had been measured simultaneously with phytoplankton sampling at the same stations and are available in the database. The sampling procedure was also described additionally, and the reference to the description of the methods used for TN measurement was provided. Following the Reviewer’s recommendation, we added two paragraphs on methodology in lines 144-156 on page 4 of the revised manuscript.

Meanwhile, we omitted the details of the entire monitoring program and full description of all methods used for measurement of the abiotic parameters that were not described and analyzed in this study. Please, also see our response to the previous comment (# 1 above).

Round 2

Reviewer 1 Report

Comments and Suggestions for Authors

Thank you to the authors for their great revisions. The manuscript is greatly improved.